# Parental Distress and Affective Perception of Hospital Environment after a Pictorial Intervention in a Neonatal Intensive Care Unit

**DOI:** 10.3390/ijerph19158893

**Published:** 2022-07-22

**Authors:** Erica Neri, Federica Genova, Marcello Stella, Alessandra Provera, Augusto Biasini, Francesca Agostini

**Affiliations:** 1Department of Psychology “Renzo Canestrari”, University of Bologna, 40127 Bologna, Italy; federica.genova@unibo.it (F.G.); alessandra.provera3@unibo.it (A.P.); f.agostini@unibo.it (F.A.); 2Pediatric and Neonatal Intensive Care Unit, Maurizio Bufalini Hospital, 47521 Cesena, Italy; marcello.stella@auslromagna.it; 3Donor Human Milk Bank Italian Association (AIBLUD), 20126 Milan, Italy; augustoclimb@gmail.com

**Keywords:** pictorial intervention, hospital environment, NICU, preterm birth, parental distress, affective perception of environment

## Abstract

Pictorial humanization is a useful intervention for the improvement of hospitalized patients’ affective states. Despite benefits in many hospital wards having been well documented, so far, no attention was paid to the Neonatal Intensive Care Unit (NICU). The aim of the present study was to evaluate the levels of distress and the affective perception of the environment experienced by parents of infants hospitalized in a NICU after the implementation of an intervention of pictorial humanization. A sample of 48 parents was recruited, 25 before the intervention was performed (Control Group), and 23 after its implementation (Pictorial Humanization Group). All parents completed the “Rapid Stress Assessment Scale” and “Scales of the Affective Quality Attributed to Place” questionnaires. Despite results showing no significant differences on parental distress, after implementation of pictorial intervention parents reported a perception of the NICU as significantly more pleasant, exciting, and arousing, and less distressing, unpleasant, gloomy, and sleepy. A higher level of distress and a perception of the environment as less relaxing was predicted for the Control Group condition. The present study suggests that the pictorial intervention represents a useful technique to create more welcoming hospital environments and to reduce the negative effects associated with infant hospitalization.

## 1. Introduction

The hospitalization of a baby in a Neonatal Intensive Care Unit (NICU) represents an unanticipated and highly stressful event not only for the infant but also for his/her parents [1,2,3,4]. The literature has deeply underlined how these parents usually report feelings of loss and uncertainty related to the high risk of serious injury, concern for short and long-term health and development, or even the baby’s death [5,6,7,8]. During hospitalization, parents can often also experience a sense of helplessness, reinforced by the need to delegate the baby’s care to medical staff [9]. Furthermore, the hospital itself can represent a stressor for parents, as the NICU rules (e.g., scheduled visiting hours) and environmental characteristics (such as noises, lights, alarms) could be perceived as unfamiliar and disturbing [10,11,12].

Therefore, parents might show elevated levels of distress, in fact, the literature shows an increased risk of depressive and anxious symptoms compared to parents with healthy children and to parents with children admitted to general pediatric wards [3,11,13,14,15,16,17]. The intensity of parental distress could be particularly severe, leading to the risk of developing post-traumatic stress symptoms [18,19] that could persist for a long time after discharge [20,21].

Parental distress not only impairs parents’ affective status but can adversely affect the quality of parenting [22,23] and the relationship with the infant [6,24], increasing the risk of long-term consequences for child development [7,25,26,27,28,29]. These consequences could be more severe in cases of higher child vulnerability, such as preterm birth. Therefore, the implementation of interventions aimed at supporting the parental role during infant hospitalization is fundamental [30,31,32]. To this end, many interventions may be listed, including educational approaches, psychological support, and psychotherapeutic strategies [33,34,35]. In recent years, many reviews and meta-analyses investigated the quality of these interventions, finding mixed results; while some programs showed beneficial effects on reducing parental distress, others showed only low or moderate positive effects [30,34,36].

Given the complexity of parental experience, the use of different approaches of intervention in combination is recommended [34]. Among these, a relevant kind of program aims to improve hospital environments, making them more suitable for young patients and their families. These interventions aim at promoting patients’ well-being and not just at reducing the levels of discomfort, consistently with the broader definition of health [37].

From this perspective, an important role is played by the intervention of humanization in healthcare environments. According to Environmental Psychology, the physical environment in which a patient receives care could play a significant role in their healing process [38,39]. In this sense, acting on the spatial, physical, and functional characteristics of the places of care, this intervention may positively influence the patients’ well-being and quality of life, reducing environmental stressors and the negative effects of hospitalization [40,41,42,43,44]. The literature has recognized the beneficial effect of specific architectural elements, such as visual (i.e., natural light, intensity of illumination), acoustic (i.e., noise reduction, therapeutic sounds, music), and olfactory characteristics (i.e., improvement of air quality) [39,40,45,46,47,48,49]. Particularly, the use of artwork or colored and pictorial installations has demonstrated positive effects on clinical indicators of health, reducing blood pressure, heart rate, pain threshold, and length of hospitalization [40,50,51,52]. Among these, the presence of nature paintings and prints in particular was associated with an improvement in patients’ conditions [50,51].

Regarding this, interventions based on the so-called “pictorial humanization” represent useful elements to elicit calmness and to decrease anxiety, agitation, and distress [39,52,53]. These interventions consist of installations of murals or panels depicting natural or artistic images. The themes of pictorial intervention are always studied with great sensitivity according to the specific patients to whom they are addressed, in order to trigger a psycho-sensorial reaction that, integrating with the rest of the environment, could favor a reassuring and restorative experience [54,55]. In a hospital context, the beneficial effect of this intervention on patients’ well-being could be associated not only with a distraction effect, but also with the activation of the “cognitive refocusing” coping strategy [56,57,58]. Indeed, the presence of a painting may sustain patients’ attention and interest, helping them to feel more comfortable and familiar in an unknown place [59,60]. Furthermore, themes of painting representing natural landscapes and of bright colors could reduce emotional states such as worrying, anxiety, and agitation [61,62,63,64,65].

Many studies have stated the positive effects of pictorial humanization in increasing well-being of patients admitted to different wards [39,42], such as Psychiatry [65], Radiology [66,67], Stem Cell Transplant Centre [68], and Pediatrics [52,54,69,70,71,72]. Referring to the latter, most of the studies directly assessed the impact of intervention on children [52,69,71,72], while only two studies [54,70] investigated the impact on hospitalized children’s parents. Specifically, both studies assessed parental “affective perception of the environment”, a construct defined by Russell and Pratt [73] to describe the emotional qualities that people attributes to the places that could influences subsequent relations with the environment [73,74]. In pediatric wards, pictorial intervention had a significant positive effect on affective perception of hospital environment [54,70], emphasizing the relevance of these interventions to support family adjustment during child hospitalization. Based on these promising findings, pictorial humanization could represent a useful intervention in the NICU environment, too, reducing the sense of unfamiliarity experienced by infants’ parents and improving their level of distress and the affective perception of the environment. Nevertheless, to our knowledge, until now, no studies have investigated the possible effect of pictorial humanization in the NICU on parental distress and affective states.

The general aim of the study was to investigate the level of distress in parents of infants admitted to a hospital NICU, before and after an intervention of pictorial humanization. Specifically, our objectives were:(a)to investigate the effect of the humanization intervention on the levels of parental distress. We hypothesized that the parents of a hospitalized infant would show lower levels of distress after pictorial humanization intervention than those observed before its implementation;(b)to evaluate the effect of the pictorial humanization intervention on the affective perception of the environment, assuming more positive responses in the group of parents evaluated after the implementation of the intervention;(c)to explore whether the dimensions of parental affective perception could be related to the level of parental distress. Indeed, according to the theoretical statement that “the first level of response to the environment is affective […] very generally governs the directions taken by subsequent relations with the environment” [74] (p. 16), an association between parental distress and affective perception of environments could be supposed. Since no previous studies have investigated this relationship, neither in the NICU nor in other hospital wards, no specific hypotheses were developed.

## 2. Materials and Methods

### 2.1. Research Design and Participants

The research design of the present study was consistent with previous studies conducted in an Italian context aimed to investigate the effect of pictorial humanization of hospital wards [54,67,70].

The total sample included 48 parents (38 mothers, 10 fathers; mean age: 33.3 ± 6.4 years; range 22–50) of infants hospitalized at the NICU of Bufalini Hospital (Ausl Romagna, Cesena, Italy).

The research consisted of two phases. In the first one, between April and July 2014, the purpose was to measure the level of parental distress and the qualities of affective perception of the non-humanized NICU environment, beforehand implementation of intervention. During this phase, the walls were white and aseptic. For this phase, 25 parents (20 mothers, 5 fathers; mean age: 32.0 ± 6.6) of infants admitted to the NICU were recruited and they represented the Control Group (CG).

At the end of this phase, a large pictorial intervention was implemented in NICU. The pictorial intervention consisted of an extensive decoration work of the corridor of the NICU: panels were affixed to all the walls, covering them entirely. Specifically, the background of the panels was blue, representing a sea landscape, and cartoon characters related to the marine theme (i.e., fish, mermaids, sailors, boats, submarines,…) were painted along all the walls. At the end of implementation, we chose to administer the same questionnaires during the same period of the year as the first administration, to avoid the law of effect.

Therefore, the second phase took place between April and July 2015, and the level of parental distress and the qualities of affective perception were assessed after the intervention of pictorial humanization. A group of 23 parents (19 mothers, 4 fathers; mean age: 34.6 ± 5.9) of infants admitted into the same Unit was recruited; this sample represented the experimental condition in the research design (Pictorial Humanization Group, PHG).

### 2.2. Procedure and Measures

All the parents were recruited during infant hospitalization at the NICU. One day a week, a psychologist went to visit parents, asking them to take part in the study. If they agreed, the psychologist gave them the informed consent and asked them to stay for 5 min in the area involved into pictorial humanization intervention. After this, they were asked to fill out two questionnaires to measure their level of distress and affective perception of the NICU environment.

The psychologist firstly asked the parents to fill out a demographic form (including the parent’s gender, age, civil status and parity, and the infant’s gender, gestational age, actual weight, twinning, the duration, and the reason for hospitalization).

To assess the level of distress, parents completed the Rapid Stress Assessment Scale (VRS [75]), a questionnaire created for the self-evaluation and perception of one’s own possible stressful reactions. VRS is based on the stress model proposed by Lazarus and Folkman [76], and evaluates the responses to stress in different psychopathologic dimensions; it includes 15 items, subdivided into 5 subscales: Anxiety, Depression, Somatization, Aggressiveness, and Lack of social support. Each item is scored on a 4-point scale (ranging from 0 to 3), and the total score ranges between 0 and 45, where lower scores describe no or low levels of distress, while higher ones indicating high stressful reactions. For each subscale it is possible to obtain a total score. VRS was used in studies on clinical and non-clinical Italian population [13,67,77,78]. In the present study, it was considered suitable to assess the level of possible parental distress as a state-measure in a hospital context, according to a previous study [13].

To assess the parental affective perception of environment, the Italian version of the “Scale of the Affective Quality Attributed to Place” (QAL [73,79,80,81]) was administered. The questionnaire is based on the circumflex model of affective quality attributed to places [73]. This measure consists of 48 adjectives, divided into 8 domains (each one consisting of 6 items), creating 4 bipolar dimensions: relaxing–distressing, exciting–gloomy, pleasant–unpleasant, and arousing–sleepy. Each item is scored on a 7-point rating scale, indicating to what extent each adjective is adequate in describing the target place (0 = not at all appropriate; 6 = completely appropriate). QAL was used in previous studies to assess the affective perceptions of hospital environment [54,67,70]. In the present study we administered the validated version by Perugini et al. [79], that that demonstrated a structural similarity to Russell’s model [73].

### 2.3. Data Analyses

Descriptive analyses (independent t-test and Pearson’s Chi-square) were conducted to investigate the homogeneity of the two groups (CG and PHG) relating to the main demographic variables for parents and infants.

We then explored whether the level of distress and parental affective perception of the environment differed between CG and PHG. Therefore, we employed two multilevel-ANOVA, considering VRS and QAL scores, respectively, as dependent variables.

Finally, we used a linear regression model with VRS total score as dependent variable and both group condition (CG versus PHG) and QAL scales as possible predictors. Given the small sample size, we conduct two separate analyses considering: firstly, group condition and QAL positive scales (relaxing, exciting, pleasant, arousing) as predictors; secondly, a model including group condition and QAL negative scales (distressing, gloomy, unpleasant, sleepy). Given that a small number of observations leads to a risk of overfitting the model [82,83,84], we counteracted the negative effects of the small sample size by implementing a bootstrap procedure. We initially performed the analysis in the estimation sample by entering all potential predictors and replicated 2000 times using bootstrap resampling [85]. The final enter model was implemented with VRS total scores determined from the bootstrap process.

Data were analyzed using SPSS for Windows version 22.0 (IBM, Armonk, NY, USA). A *p* value < 0.05 was considered statistically significant.

## 3. Results

### 3.1. Descriptive Characteristics

Preliminary analyses showed that the two groups were homogeneous in relation to all parent and infant variables (Table 1).

### 3.2. Effect of the Intervention of Pictorial Humanization on Parental Distress

When we investigated parental distress, no significant differences between groups emerged in any scale of VRS (all *p* values > 0.05) (Table 2).

### 3.3. Effect of the Intervention of Pictorial Humanization on Affective Perception of the NICU Environment

Regarding the affective perception of the NICU environment, significant differences emerged between the two groups in each subscale. Particularly, PHG group showed significantly higher scores than did the CG in three of the four positive scales: pleasant (F(1,45) = 12.367; *p* = 0.001), exciting (F(1,45) = 16.419; *p* < 0.0005) and arousing (F(1,45) = 6.317; *p* = 0.016) (Figure 1).

Conversely, no significant differences between CG and PHG were found on relaxing scale (F(1,45) = 0.289; *p* = 0.594) (Figure 1).

Furthermore, PHG obtained significantly lower scores than did the CG in all the negative scales: distressing (F(1,45) = 4.355; *p* = 0.043), unpleasant (F(1,45) = 9.078; *p* = 0.004), gloomy (F(1,45) = 10.808; *p* = 0.002) and sleepy (F(1,45) = 7.073; *p* = 0.011) (Figure 2).

### 3.4. Potential Predictors of Parental Distress

As mentioned above, two separate regression analyses were conducted to explore the role of group condition and affective perception of environment on parental distress.

The first model predicted the level of distress considering group condition and the four positive scales at QAL (Table 3). A statistically significant regression equation was found (F(5,44) = 6.523, *p* < 0.0005), with an R^2^Adjusted = 0.386. According to the model, group condition and scores at relaxing subscale significantly contributed to VRS total Score (β = −0.377, t(5) = −2.422, *p* = 0.020; β = −0.847, t(5) = −4.894, *p* < 0.0005, respectively). The direction of these relationships suggested that a higher level of distress was predicted by belonging to the control group condition, namely, CG parents showed an increased probability to have higher VRS total score; a perception of the NICU environment as less relaxing (Table 3).

The second model included group condition and the four negative QAL scales and tested their prediction of VRS total score. No statistically significant equation emerged (F(5,44) = 1.525, *p* > 0.05, R^2^Adjusted = 0.056).

## 4. Discussion

The main aim of the present study was to evaluate the effects of an intervention of pictorial humanization on the levels of distress and the affective perception of the environment experienced by parents of infants hospitalized in the NICU. The research is part of a topic of growing interest in the international literature, that is, the implementation of humanization interventions of healthcare environments aimed at increasing patient well-being and reducing the negative effects of hospitalization. The literature has recognized the effects of interventions of humanization on affective perception in several hospital wards, such as Psychiatry, Radiology, Stem Cell Transplant Centre, and Pediatrics [42,52,54,65,66,67,68,69,70,71,72]. Nevertheless, no attention has been paid to the NICUs, often described by parents as unfamiliar, disturbing, and highly distressing environments [10,11,12].

According to the first aim, we analyzed the data to investigate whether the levels of distress in preterm babies’ parents could differ after the intervention of pictorial humanization. Contrary to our hypotheses, no significant differences emerged between the two groups of parents, suggesting that the intervention was not related to global levels of parental distress nor to single dimensions. A possible explanation could be related to the high scores of global distress displayed by parents; the emotional experience of being a parent after infant hospitalization in the NICU is complex and intense [1,2,3,4,17] and could conceal the potential benefits of this kind of intervention. Indeed, looking at the VRS continuous scores in comparison with the VRS scores which emerged from other studies, we found a more intense distress than that of the normative population [67,75,78,79], and of hospital patients, such as women undergoing to breast cancer screening [67] and parents of children undergoing anesthesia for day-hospital surgery [13]. Moreover, the effect of other factors should be taken into consideration; variables related to infant or parents’ condition (i.e., infant gestational age, reason of admission, age, gender), could reduce the effect of the intervention. Further studies should be recommended in order to explore the possible role of these variables.

According to the second aim, we assessed the impact of pictorial humanization intervention on parental affective perception of the NICU, finding a significant effect. Indeed, in almost all scales of QAL significant differences emerged between groups, with more positive outcomes for the PHG parents. This result is consistent with previous studies [49,67,86] and, specifically, with those regarding pediatrics patients’ parents [54,70]. In the present study we found, in particular, that the scores for all negative subscales of QAL (distressing, unpleasant, gloomy and sleepy) were significantly lower after the implementation of intervention. This result has clinical implications, suggesting that this intervention may reduce the sense of unfamiliarity and discomfort reported by parents during their stay in the NICU [10,11,12]. This explanation is further supported by the fact that parents, after the pictorial humanization, obtained higher scores at three of the positive QAL subscales: pleasant, exciting, and arousing. This finding suggests that the parents perceived the NICU environment as a more comfortable place.

Nevertheless, no differences emerged in the case of the relaxing subscale. We could suppose that this result is related to the highly distressing experience of having an infant hospitalized in the NICU. Monti et al. [70] have previously suggested that the severity of the child’s health condition could influence the effects of this kind of intervention. Indeed, despite the parents of their sample displaying significant differences in all the subscales of QAL (globally an improvement after the pictorial humanization), in the case of the exciting subscale the improvement was significant only for parents of children with less severe health conditions, compared to parents whose infants showed chronic illness. Authors suggested that, due to the high level of stress experienced by these last parents, the improvement given by the intervention could be less effective. Considering our results, this explanation could be relevant too, given that all the infants of our sample were experiencing severe health problems due to their prematurity. The difference of significance in QAL subscales of the two studies (exciting vs. relaxing) could be related to the fact the study by Monti et al. [70] regarded the affective perception of a Pediatric Unit including a wide range of child ages (up to 11 years), while our study focused on the NICU environment and included only preterm babies.

Taken together, these results seemed to suggest that the intervention of pictorial humanization is associated with the affective perception of environment but not with parental distress. If this result support the importance of creating more suitable environments for patients and their families [49,87], it should also be considered in addition to approaches focused on patient affective states, especially in contexts where hospitalization is an extremely stressful experience.

Finally, we investigated whether the dimensions of affective parental perception could predict the level of parental distress. The regression model showed that a positive affective perception of environment could significantly contribute to the global level of parental well-being; specifically, distress decreased when parents perceived a more relaxing environment. This result is consistent with previous studies [39,49,88], where guidelines for the contents of pictorial humanization suggested that this intervention should include elements able to promote quietness and restoration. Conversely, colors or themes that could highly stimulate patients were not recommended, because this could increase agitation and worsen health and emotional states [39,49,88].

Interestingly, despite no effects of the intervention on parental distress emerged when MANOVAs were employed, pictorial intervention showed a significant effect in regression analysis, suggesting that parents belonging to control their condition had a higher probability of displaying a high VRS total score. Therefore, despite the humanized environment itself being unable to directly affect parental distress, its impact could emerge only when a positive affective perception of environment was also considered.

Finally, the regression model including QAL negative scales did not reach statistical significance. This unexpected result seemed to suggest that the positive dimensions of affective perceptions of environment contribute more to parental distress than negative ones. Nevertheless, given the exploratory aim of this investigation, and the small size of our analyses, these results should be considered in further investigations in order to be confirmed.

Although promising, the present study results should be considered as preliminary, and some limitations of the study should be noted. First, the results need to be confirmed on larger samples. Despite the sample was large enough to guarantee the sensitivity of our analyses [88,89], the power of the analyses was low, and the testing of more sophisticated hypotheses was not possible [90]. Second, future studies are needed to confirm the results while also controlling for the effects of other variables. Indeed, specific characteristics of parents and infants (i.e., parental age or gender, reason of hospitalization, severity of conditions) could play a relevant role in influencing parental affective reactions [13,17,70]: therefore, all these variables need to be considered for their possible influences on the outcomes. Another limitation of this study relates to the absence of data about the specific characteristics of pictorial intervention. Previous studies [39,48,91] identified the specificity of the effects of the colors, size (single paintings vs. wall coverage) and content (simple images vs. natural landscapes) of pictorial humanization. While we were not able to collect data on the specific elements of the pictorial representations, we are aware that a future study should also investigate this issue.

Notwithstanding the limitations above, the strength of our study is the implementation of pictorial intervention in an NICU, contributing towards making it a suitable environment for fostering the activation of parental resilience rather than simply as a stressor.

Future studies and replications are, however, needed to generalize the results and support further recommendations for clinical care.

## 5. Conclusions

The present study investigated the possible benefits of pictorial humanization on parental distress and affective perception of environment during infant hospitalization in the NICU. In summary, positive effects of the pictorial intervention were found. Indeed, despite no change in parental distress emerging after implementation of the intervention, it ultimately proved to be a significant contributing factor towards parental states of mind, along with the perception of a highly relaxing environment.

Overall, these results extend previous studies on the effects of pictorial humanization, including the NICU as a hospital ward that could benefit from this intervention. The innovative aspect of our study relates to the implementation of pictorial intervention in a ward recognized as potentially traumatic for parents; coping with the infant hospitalization in the NICU means that the parents have to deal with mixed feelings, ranging from fear of death to optimism, from uncertainty and instability to trust and sense of stability. Therefore, these parents may need special support, fostering the development and the strengthening of emotional resources. More generally, these considerations could suggest the relevance of proposing this kind of intervention in several contexts where parental stress is high, beyond child hospitalization, to support parental adjustment. In this sense, the present study has potential clinical and practical implications for the care of families. The perception of the hospital environment as a cold or hostile place might increase the risk of emotional burden and psychological discomfort for parents. Conversely, a humanized environment could facilitate engagement in behaviors and emotions that support the healing process [92]. Therefore, these parents could regard themselves as being in a warmer and human-friendly environment; in other words, “a place to go, not to stay” [91] (p. 372).

## Figures and Tables

**Figure 1 ijerph-19-08893-f001:**
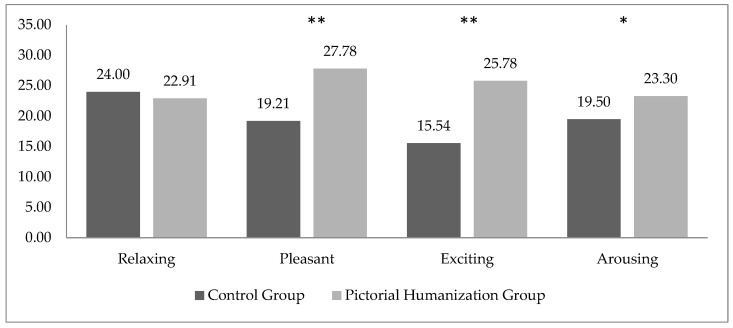
Mean scores for positive QAL scales in CG and PHG parents. * *p* < 0.05; ** *p* < 0.005.

**Figure 2 ijerph-19-08893-f002:**
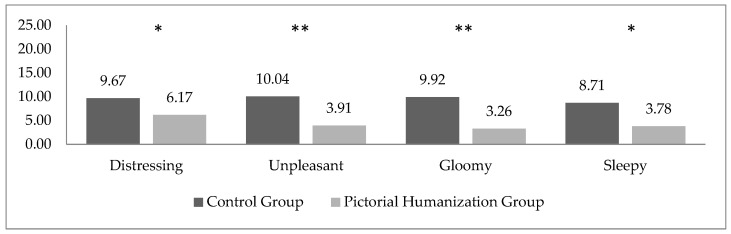
Mean scores for negative QAL scales in CG and PHG parents. * *p* < 0.05; ** *p* < 0.005.

**Table 1 ijerph-19-08893-t001:** Descriptive characteristics of sample.

	Control Group (N = 25)	Pictorial Humanization Group (N = 23)	t/X^2^	*p*
Parents				
Gender, *n* (%)			0.138	0.710
Mother	20 (80)	19 (83)		
Father	5 (20)	4 (17)		
Age, years, m ± sd	32.0 ± 6.6	34.6 ± 5.9	−1.407	0.167
Marital Status, *n* (%)			3.511	0.173
Married	13 (52)	15 (65)		
Unmarried	7 (28)	2 (9)		
Cohabitant	5 (20)	6 (26)		
Parity, *n* (%)			1.928	0.165
Primiparous	19 (76)	14 (61)		
Multiparous	6 (24)	9 (39)		
Infants				
Gender, *n* (%)			<0.0001	0.999
Female	14 (56)	13 (57)		
Male	11 (44)	10 (43)		
Gestational Age, m ± sd	35.3 ± 2.9	34.3 ± 3.6	0.897	0.375
Infant Weight, m ± sd	2639.1 ± 1642.5	1927.64 ± 845.4	1.814	0.075
Length of Hospitalization, m ± sd	19.1 ± 16.7	26.1 ± 19.9	−0.892	0.382
Twinning, *n* (%)			0.186	0.666
Yes	6 (24)	6 (26)		
No	19 (76)	17 (74)		
Reason of hospitalization, *n* (%)			4.349	0.114
Preterm Birth	18 (72)	20 (87)		
Icterus	1 (4)	2 (9)		
Other	6 (24)	1 (4)		

**Table 2 ijerph-19-08893-t002:** Mean scores for VRS scales and total score in CG and PHG parents.

	Control Group (N = 25)	Pictorial Humanization Group (N = 23)	F	*p*
VRS Total score	14.90 ± 7.65	18.65 ± 10.08	1.806	0.187
Anxiety	3.33 ± 2.08	3.40 ± 2.19	0.010	0.921
Depression	2.90 ± 2.23	4.55 ± 3.20	3.665	0.063
Somatization	3.86 ± 1.77	4.20 ± 2.26	0.294	0.591
Aggressiveness	1.90 ± 1.97	2.70 ± 2.43	1.330	0.256
Lack of social support	2.90 ± 1.67	3.80 ± 2.07	2.337	0.134

Data are mean ± standard deviation.

**Table 3 ijerph-19-08893-t003:** Regression model identifying the effect of group intervention and QAL positive subscales on VRS total score.

	T	β	t	*p*
Constant	52.604		10.584	<0.0005
Group Condition	−4.953	−0.377	−2.422	0.020
Relaxing	−0.832	−0.847	−4.894	<0.0005
Pleasant	0.236	0.316	0.889	0.379
Exciting	0.316	0.463	1.494	0.143
Arousing	−0.303	−0.242	−1.511	0.139

## Data Availability

Data are available upon request due to privacy restrictions.

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
