# Peer review of "Parental Distress and Affective Perception of Hospital Environment after a Pictorial Intervention in a Neonatal Intensive Care Unit"

_ijerph, 2022, doi:10.3390/ijerph19158893_

Round 1

Reviewer 1 Report

Congratulations on the work done. I think it is a very important topic for further study.

Author Response

We thank Reviewer for his/her positive comments.

Reviewer 2 Report

Interesting study but in my opinion lacks operational definitions of pictorial humanization as well as affective perception. This makes it very difficult to quantify the data to be gathered. It needs a lot of work in describing in detail the specifics of the parents that partcipated in the study

Author Response

We thank the Reviewer for the attention paid to our work. We agree with the critical points suggested. So, we revised introduction section, adding theoretical definition of principal constructs (pictorial humanization and affective perception of place (p. 2; lines: 80-86; p. 2; lines: .92-97, respectively). We improved the method section, better explaining the distinction of the two phases and when assessments were run, the kind of pictorial intervention and the recruitment (p. 3, lines: 132-138). We hope that this will improve the understanding of the experimental design and support the replication of results. 

Reviewer 3 Report

The present study aimed at evaluating the effects of an intervention of pictorial humanization on the levels of distress and the affective perception of the environment experienced by 48 parents of infants hospitalized in NICU. 

Overall, I found the paper well-structured. I have only a minor comment: considering the small sample and the cross-sectional nature of the study that prevent to taken into account causality, Authors should replace the term "influence" throughout the manuscript.

Author Response

We thank the Reviewer for his/her comments. We agree that the small sample size could suggest an association among variables rather than a causal relationship. We revised manuscript and changed terms in title, abstract (pag 1, lines 16-18,22-23), introduction (pag 3 lines 114), result (pag 5. Lines 199, 205; p.6 lines: 226) and discussion (pag 7 lines 249-250, 281; pag 8 lines 295,317; pag 9 lines 345,347), in order to reduce the relationship of causality.

Round 2

Reviewer 2 Report

The understanding of the information in the manuscript is improved. However, what can not be addressed are the methodological issues in identifying and quantifying such a subjective process as Pictorial Humanization. The methodology is where, in my opinion, the major problem is. How do you measure “the affective perception of the environment…”? How do you do realizable statistical analysis with such vague constructs and such a small sample size in one setting. The operational definition of “pictorial humanization” is not clearly nor objectively provided. The intervention is not clearly defined and quantified. It is the methodological problems and the vague subject matter that is difficult to objectively quantified that makes this very interesting approach to evaluating stress in the NICU and possible intervention worthwhile. However, in my opinion, at this time there is a need to focus on developing a methodology that quantifies what is to be measured and increase the sample size.

Author Response

We thank reviewer for his/her considerations. We read them carefully and revised manuscripts including sentences to improve the understanding of methodology and theoretical background. Find attach all or revisions.
